# Task-Based Top-Down Modulation Network for Multi-Task-Learning Applications

## Abstract

A general problem that received considerable recent attention is how to perform multiple tasks in the same network, maximizing both efficiency and prediction accuracy. A popular approach consists of a multi-branch architecture on top of a shared backbone, jointly trained on a weighted sum of losses. However, in many cases, the shared representation results in non-optimal performance, mainly due to an interference between conflicting gradients of uncorrelated tasks. Recent approaches address this problem by a channel-wise modulation of the feature-maps along the shared backbone, with task specific vectors, manually or dynamically tuned. Taking this approach a step further, we propose a novel architecture which modulate the recognition network channel-wise, as well as spatial-wise, with an efficient top-down image-dependent computation scheme. Our architecture uses no task-specific branches, nor task specific modules. Instead, it uses a top-down modulation network that is shared between all of the tasks. We show the effectiveness of our scheme by achieving on par or better results than alternative approaches on both correlated and uncorrelated sets of tasks. We also demonstrate our advantages in terms of model size, the addition of novel tasks and interpretability.
Code will be released.

## 1 Introduction

The goal of multi-task learning is to improve the learning efficiency and increase the prediction accuracy of multiple tasks learned and performed together in a shared network.

Over the years, several types of architectures have been proposed to combine multiple tasks training and evaluation. Most current schemes assume task-specific branches, on top of a shared backbone (Figure 1a) and use a weighted sum of tasks losses, fixed or dynamically tuned, to train them (Chen et al., 2017; Kendall et al., 2018; Sener & Koltun, 2018). Having a shared representation is more efficient from the standpoint of memory and sample complexity and can also be beneficial in cases where the tasks are correlated to each other (Maninis et al., 2019). However, in many other cases, the shared representation can also result in worse performance due to the limited capacity of the shared backbone and interference between conflicting gradients of uncorrelated tasks (Zhao et al., 2018). The performance of the multi-branch architecture is highly dependent on the relative losses weights and the task correlations, and cannot be easily determined without a "trial and error" phase search (Kendall et al., 2018).

Another type of architecture (Maninis et al., 2019) that has been recently proposed uses task specific modules, integrated along a feed-forward backbone and producing task-specific vectors to modulate the feature-maps along it (Figure 1b). Here, both training and evaluation use a single tasking paradigm: executing one task at a time, rather than getting all the task responses in a single forward pass of the network. A possible disadvantage of using task-specific modules and of using a fixed number of branches, is that it may become difficult to add additional tasks at a later time during the system life-time. Modulation-based architectures have been also proposed by Strezoski et al. (2019) and Zhao et al. (2018) (Figure 1c). However, all of these works modulate the recognition network channel-wise, using the same modulation vector for all the spatial dimension of the feature-maps.

We propose a new type of architecture with no branching, which performs single task at a time but with no task-specific modules (Figure 1d). The core component of our approach is a top-down (TD)

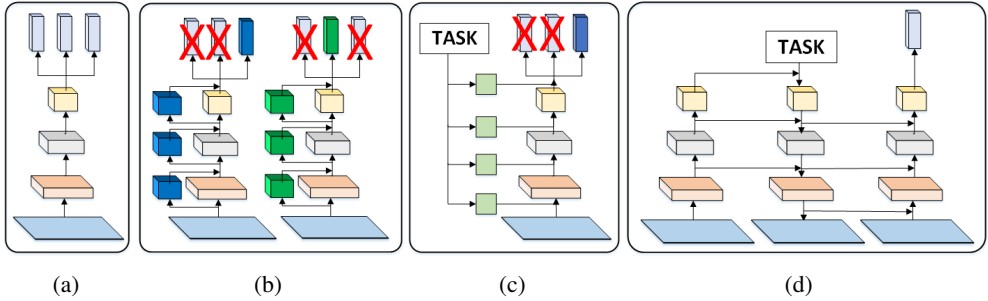

Figure 1: (a) Multi branched architecture, task specific branches on a top of a shared backbone, induces capacity and destructive interference problems, force careful tuning. Recently proposed architectures: (b) using tasks specific modules and (c) using channel-wise modulation modules. (d) **Our architecture:** a top-down image-aware full tensor modulation network with no task specific modules.

modulation network, which carries the task information in combination with the image information, obtained from a first bottom-up (BU1) network, and modulates a second bottom-up (BU2) network common for all the tasks. In our approach, the modulation is channel-wise as well as spatial-wise (a full tensor modulation), calculated sequentially along the TD stream. This allows us, for example, to modulate only specific spatial locations of the image depending on the current task, and get interpretability properties by visualizing the activations in the lowest feature-map of the TD stream. In contrast to previous works, our modulation mechanism is also "image-aware" in the sense that information from the image, extracted by the BU1 stream, is accumulated by the TD stream, and affects the modulation process.

The main differences between our approach and previous approaches are the following: First, as mentioned, our approach does not use multiple branches or task-specific modules. We can scale the number of tasks with no additional layers. Second, our modulation scheme includes a spatial component, which allows attention to specific locations in the image, as illustrated in figure 2a for the Multi-MNIST tasks (Sabour et al., 2017). Third, the modulation in our scheme is also image dependent and can modulate regions of the image based on their content rather than location (relevant examples are demonstrated in figures 2b and 2c).

We empirically evaluated the proposed approach on three different datasets. First, we demonstrated on par accuracies with the single task baseline on an uncorrelated set of tasks with MultiMNIST while using less parameters. Second, we examined the case of correlated tasks and outperformed all baselines on the CLEVR (Johnson et al., 2017) dataset. Third, we scaled the number of tasks and demonstrated our inherent attention mechanism on the CUB200 (Welinder et al., 2010) dataset. The choice of datasets includes cases where the tasks are uncorrelated (Multi-MNIST) and cases where the tasks are relatively correlated (CLEVR and CUB200). The results demonstrate that our proposed scheme can successfully handle both cases and shows distinct advantages over the channel-wise modulation approach.

## 2 RELATED WORK

Our work draw ideas from the following research lines:

**Multiple Task Learning (MTL)**   Multi task learning has been used in machine learning well before the revival of deep networks (Caruana, 1997). The success of deep neural networks in single task performance (e.g. in classification, detection and segmentation) has renewed the interests of the computer vision community in the field (Kokkinos, 2017; He et al., 2017; Redmon & Farhadi, 2017). Although our primary application area is computer vision, multi task learning has also many application in other fields like natural language processing (Hashimoto et al., 2016; Collobert & Weston, 2008) and even across modalities (Bilen & Vedaldi, 2016). We further refer the interested reader to a review that summarizes recent work in the field (Ruder, 2017).

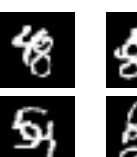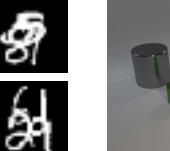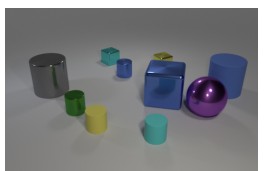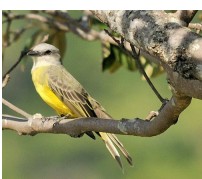

(a) M-MNIST examples     (b) CLEVR example     (c) CUB200 example

Figure 2: Images examples with their corresponding tasks. Our architecture benefits from its built-in image-aware, task dependent, localized attention mechanism. (a) M-MNIST examples, the tasks are to recognize the digits by their location. (b) CLEVR example, an example task is to determine whether there is a sphere to the right of a cylinder. (c) CUB200 example, an example task is to classify the bird's neck color.

Over the years, several types of architectures have been proposed in computer vision to combine the training and evaluation of multiple tasks. First works used several duplications (as many as the tasks) of the base network, with connections between them to pass useful information between the tasks (Misra et al., 2016; Rusu et al., 2016). These works do not share computations and cannot scale with the tasks. More recent architectures, which are in common practice these days, assume task-specific branches on a top of a shared backbone, and use a weighted sum of losses to train them. The joint learning of several tasks has proven to be beneficial in several cases (He et al., 2017) but can also decrease the results of some of the tasks due to a limited network capacity, uncorrelated gradients from the different tasks (sometimes called destructive inference) and different learning rates (Kirillov et al., 2019). A naive implementation of multi-task learning requires careful calibration of relative losses of the different tasks. To address these problem several methods have been proposed: "Grad norm" (Chen et al., 2017) dynamically tunes gradients magnitudes over time, to obtain similar learning rates of the different tasks. Kendall et al. (2018) uses a joint likelihood formulation to derive task weights based on the intrinsic uncertainty in each task. Sener & Koltun (2018) applies an adaptive weighting of the different tasks, to force a pareto optimal solution to the multi task problem.

Along an orthogonal line of research, other works suggested to add task-specific modules to be activated or deactivated during training and evaluation, depending on the task at hand. Liu et al. (2019b) suggests task specific attention networks in parallel to a shared recognition network. Maninis et al. (2019) suggests adding several types of low-weight task-specific modules (e.g. residual convolutional layers, squeeze and excitation (SE) blocks and batch normalization layers) along the recognition network. Note that the SE block essentially creates a modulation vector, to be channel-wise multiplied with a feature-map. Modulation vectors have been further used in Strezoski et al. (2019) for a recognition application, in Cheung et al. (2019) for continual learning applications and in Zhao et al. (2018) for a retrieval application and proved to decrease the destructive interference between tasks and the effect of the catastrophic forgetting phenomena.

Our design, in contrast, does not use multi-branch architecture, nor task-specific modules. Our network is fully-shared between the different tasks. Compared to Zhao et al. (2018), we modulate the feature-maps in the recognition network channel-wise as well as spatial-wise, depending on both the task and the specific image at hand.

**Top-Down Modulation Networks** Neuroscience research provides evidence for a top-down context, feedback and lateral processing in the primate visual pathway (Gazzaley & Nobre, 2012; Gilbert & Sigman, 2007; Lamme et al., 1998; Hopfinger et al., 2000; Piëch et al., 2013; Zanto et al., 2010) where top-down signals modulate the neural activity of neurons in lower-order sensory or motor areas based on the current goals. This may involve enhancement of task-relevant representations or suppression for task-irrelevant representations. This mechanism underlies humans ability to focus attention on task-relevant stimuli and ignore irrelevant distractions (Hopfinger et al., 2000; Piëch et al., 2013; Zanto et al., 2010).

In this work, consistent with this general scheme, we suggest a model that uses top-down modulation in the scope of multi-task learning. Top down modulation networks with feedback, implemented as conv-nets, have been suggested by the computer vision community for some high level tasks (e.g.

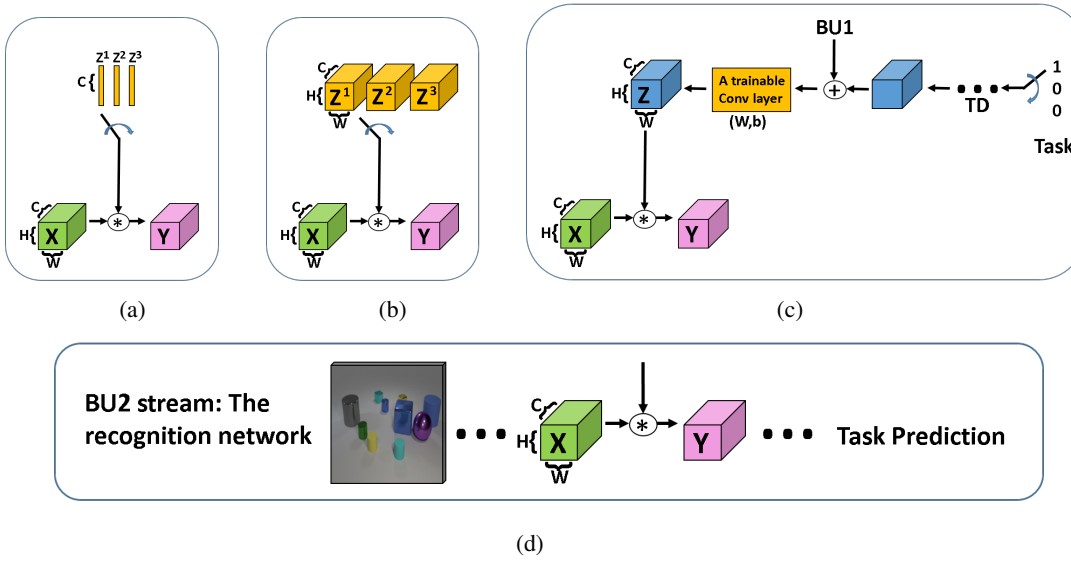

Figure 3: **Several types of modulation modules**, the trainable parameters are illustrated in yellow. (d) For simplicity we show only one modulation stage, where **X** is the input tensor to be modulated and **Y** is the output tensor. (a) task-dependent vector-modulation architecture (Zhao et al., 2018), the modulation vectors are switched by the task and explicitly being optimized. (b) Hypothetical extension of (a) to spatial-wise modulation tensors, cannot be done in practice due to the huge number of parameters to optimize. (c) **Our approach:** We optimize the parameters in the convolutional layers along the top-down network and use the created featuremaps as the modulation tensors.

re-classification (Cao et al., 2015), keypoints detection (Carreira et al., 2016; Newell et al., 2016), crowd counting (Sam & Babu, 2018), curriculum learning (Zamir et al., 2017) etc.) and here we apply them to multi-task learning applications.

## 3 APPROACH

We will first describe the task-dependent vector-modulation mechanism, as proposed in Zhao et al. (2018), illustrated in figure 3a, and then describe our architecture (figure 3c) in detail.

### 3.1 VECTOR-MODULATION MODULE

A vector-modulation (a channel-wise modulation) of a given tensor **X** by a vector $z$ is defined as the product of the elements of the vector $z$ and the corresponding channels of the tensor **X**. Each element in the output tensor **Y** is calculated by:

$$\mathbf{Y}(y, x, ch) = \mathbf{X}(y, x, ch) * \mathbf{z}(ch) \tag{1}$$

where **X** is the tensor to be modulated and **Y** is the modulated tensor, both in the form $(H \times W \times C)$ where H, W, are the spatial dimensions of the tensors and C their channel dimension. The vector $z \in \mathbb{R}^C$ has dimension equal to the number of channels of **X**, **Y**. $x, y, ch$ are the column, row, and channel number and indicate a specific element in a tensor. In training the network, the elements of $z$ are considered as parameters and are directly optimized ($C$ additional parameters).

In the scope of Multi Task Learning, where several tasks co-exist, the network switches on the fly between several modulation vectors $\left\{ \mathbf{z}^k \right\}_{k=1}^K$ where $K$ is the number of tasks, see figure 3a for illustration. The network performs one task at a time, and the modulated tensor **Y** depends on the selected task. This vector-modulation module has been used in Zhao et al. (2018) separately for every stage in the recognition network (with additional $CK$ parameters in every stage).

Two limitations of this module are that it ignores the spatial dimensions of the image, and the lack of information from the image itself. The possible use of the same strategy to explicitly optimize spatial-aware modulation tensors $\left\{ \mathbf{Z}^k(y, x, ch) \right\}_{k=1}^{K}$ (figure 3b) was discussed in Zhao et al. (2018) but was deemed infeasible due to the large amount of added parameters (HWCK additional parameters in every stage). Our method addresses both of these issues in an efficient manner and demonstrates better accuracy, showing that spatial-wise modulation and the use of image information are beneficial to many kinds of tasks.

## 3.2 TOP-DOWN TENSOR-MODULATION MODULE

A tensor modulation is defined by:

$$\mathbf{Y}(y, x, ch) = \mathbf{X}(y, x, ch) * \mathbf{Z}(y, x, ch) \tag{2}$$

Where $\mathbf{Z} \in \mathbb{R}^{H \times W \times C}$ is a modulation tensor.

To avoid the infeasible computation of directly optimizing $\mathbf{Z}$, we propose the use of created featuremaps as the modulation tensors. Practically, we use a dedicated top-down (TD) convolutional stream, shared between the tasks, to create the modulation featuremaps, and optimize the weights of the convolutional layers instead of directly optimizing the modulation tensors (figure 3c). The number of added parameters in this case depends on the precise architecture of the TD stream but can be approximately estimated by $3 \times 3 \times C^2$ parameters for each convolutional layer (several convolutional layers may be used in one stage). Avoiding the dependency of the number of added parameters on H, W and K allows us to apply the proposed architecture to large images and to scale the number of tasks, as illustrated our experiments.

**A gated modulation module with a residual connection**   We further define a gated modulation module with a residual connection as:

$$\mathbf{Y}(y, x, ch) = \mathbf{X}(y, x, ch) + \mathbf{X}(y, x, ch) * \sigma(\mathbf{Z}(y, x, ch)) = \mathbf{X}(y, x, ch) \otimes \mathbf{Z}(y, x, ch) \tag{3}$$

where the modulation tensor $\mathbf{Z}$ is gated with a *sigmoid* or a *tanh* function before the multiplication and then added to the input tensor $\mathbf{X}$ through a residual connection. The residual gated modulation with $\mathbf{Z}$ is equivalent to the modulation with $\tilde{\mathbf{Z}} = (1 + \sigma(\mathbf{Z}))$. Motivated by our ablation studies 4.3.2, unless stated otherwise, we use the gated modulation as defined in Eq. 3 in all of our experiments. For simplicity we denote this operation by the symbol $\otimes$.

## 3.3 IMAGE-AWARE TASK-DEPENDENT TOP-DOWN MODULATION NETWORK

An illustration of our network design is shown in figure 1d. In our design, a bottom-up (BU2) recognition network is modulated by a top-down (TD) modulation stream. The inputs to the TD stream are the current task $k$, and the feature-maps along the first bottom-up stream (BU1, where BU1 and BU2 share the same weights), added to the TD stream via lateral connections. The outputs of the TD stream are its feature-maps, which sequentially modulate the tensors along the recognition network (BU2). Figure 3c illustrate our architecture for one modulation step.

**Auxiliary losses**   Our architecture can be naturally decomposed into three sub-networks (BU1, TD, BU2), allowing the structural advantage of adding auxiliary losses at the end of the BU1 or TD streams. This possibility is application-dependent. In the scope of multi-task learning, the TD auxiliary loss might be beneficial because it allows the use of spatial information in a task-dependent manner. This issue is further discussed in section 4.3.4 where we demonstrate the use of a localization loss in the last TD featuremap. Applying the localization loss in train time allows us to obtain an attention map in inference time, which illustrates the relative weights assigned by the network to different locations in the image.

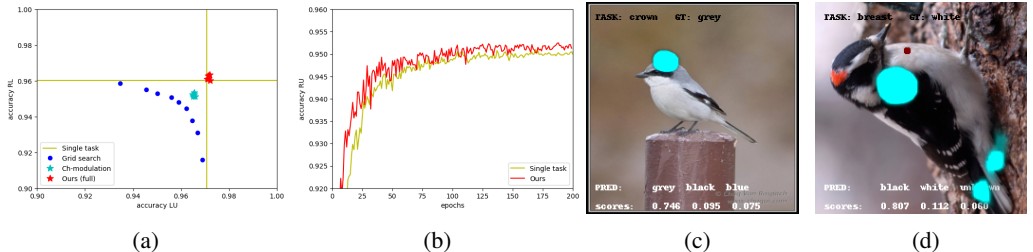

(a)     (b)     (c)     (d)

Figure 4: (a) Multi-MNIST accuracy scatter plot, top-right is better. We show higher accuracies than the single-task baseline while using much less parameters (b) Multi-MNIST: Typical training curves, we show better accuracy than the single-task baseline along the training (c), (d) A close look into the network decision making process; (b) good example: the crown area is well localized and the prediction follows the ground truth, (c) an error example: the breast isn't localized well enough.

# 4 DATABASES AND EXPERIMENTS

## 4.1 DATABASES

We validate our approch on three different datasets:

**MultiMNIST** MultiMNIST (Sabour et al., 2017) is a multi-task learning version of the MNIST dataset in which multiple MNIST images are placed on the same image. We use 2, 3 and 4 classes experiments built as suggested by Sener & Koltun (2018). Several examples are demonstrated in Figure 2a. In the 2-classes experiment the tasks are: classifying the digit on the top-left (task-LU) and classifying the digit on the bottom-right (task-RL). We corespondently add (task-LL) and (task-RU) for classifying the digits on the bottom-left and top-right on the 3 and 4-classes experiments. The digits are independently chosen and the tasks are considered to be uncorrelated. We use 60K examples and directly apply LeNet (LeCun et al., 1998) as the underlying backbone in our experiments.

**CLEVR** CLEVR is a synthetic dataset, consists of 70K training images and 15K validation images, mainly used as a diagnostic dataset for VQA. The dataset includes images of 3D primitives, with multiple attributes (shape, size, color and material) and a set of corresponding (question-answer) tuples. We followed the work Liu et al. (2019a), which suggested to use CLEVR not as a VQA dataset, but rather as a referring expression dataset, and further adapt it to a multi-task learning methodology. The tasks in our setup consist of 4 questions ("Are there exactly two cylinders in the image?", "Is there a cube right to a sphere?", "Is there a red sphere?" and "Is the leftmost sphere in the image large?"), arbitrarily chosen, with various compositionary properties.

**CUB200** is a fine grained recognition dataset that provides 11,788 bird images (equally divided for training and testing) over 200 bird species with 312 binary attribute annotations, most of them referring to the colors of specific birds' parts. In contrast to other work (Strezoski et al., 2019) that used all of the 312 attributes as a yes/no question, we re-organized the attributes as a multi-task problem of 12 tasks (for 12 annotated bird's parts) each with 16 classes (the annotated colors + an unknown class) and train using a multi-class cross-entropy loss. To demonstrate our interpretability capability, we further used the parts' location, annotated by a single point to each seen part, as an auxiliary target at the end of the TD stream.

## 4.2 EXPERIMENTS

**architecture** We use LeNet, VGG-11 and resnet-18 as our backbone BU architectures for the Multi-MNIST, CLEVR and CUB-200 experiments correspondingly. Each of the backbones has been divided to two parts; a first part that consists mainly of the convolutional layers of the backbone and a second part with the fully connected layers (the classifier).

Table 1: Performance (mean $\pm$ std of 5 repetitions) on Multi-MNIST, uncorrelated tasks, higher is better. Our architecture achieves consistently better accuracies than the single task baseline while using much less parameters (the third column shows the ratio between the number of parameters and a standard LeNet architecture). Scaling the number of tasks in our architecture costs no additional hardware.

| | ALG | #$P$ | LU accuracy | RL accuracy | LL accuracy | RU accuracy |
|---|---|---|---|---|---|---|
| 2 tasks | Single task | x2 | 96.99 $_{\pm 0.07}$ | 95.93 $_{\pm 0.10}$ | | |
| | Uniform scaling | x1.12 | 95.86 $_{\pm 0.08}$ | 94.75 $_{\pm 0.16}$ | | |
| | MOO | x1.12 | 96.25 $_{\pm 0.34}$ | 95.38 $_{\pm 0.16}$ | | |
| | ch-mod | x1.002 | 96.53 $_{\pm 0.04}$ | 95.21 $_{\pm 0.09}$ | | |
| | Ours | x1.29 | **97.16** $_{\pm 0.05}$ | **96.19** $_{\pm 0.14}$ | | |
| 3 tasks | Single task | x3 | **90.84** $_{\pm 0.03}$ | 90.64 $_{\pm 0.03}$ | 83.22 $_{\pm 0.06}$ | |
| | Uniform scaling | x1.25 | 87.92 $_{\pm 0.25}$ | 87.55 $_{\pm 0.29}$ | 78.36 $_{\pm 0.67}$ | |
| | MOO | x1.25 | 87.38 $_{\pm 0.31}$ | 86.82 $_{\pm 0.36}$ | 79.42 $_{\pm 0.44}$ | |
| | ch-mod | x1.005 | 88.42 $_{\pm 0.53}$ | 88.70 $_{\pm 0.27}$ | 80.27 $_{\pm 0.42}$ | |
| | Ours | x1.31 | 90.83 $_{\pm 0.18}$ | **90.67** $_{\pm 0.24}$ | **83.45** $_{\pm 0.38}$ | |
| 4 tasks | Single task | x4 | 95.73 $_{\pm 0.14}$ | 92.97 $_{\pm 0.08}$ | 93.11 $_{\pm 0.21}$ | 94.81 $_{\pm 0.11}$ |
| | Uniform scaling | x1.37 | 92.86 $_{\pm 0.30}$ | 89.30 $_{\pm 0.57}$ | 88.88 $_{\pm 0.69}$ | 91.80 $_{\pm 0.32}$ |
| | MOO | x1.37 | 92.96 $_{\pm 0.15}$ | 90.04 $_{\pm 0.46}$ | 89.87 $_{\pm 0.26}$ | 92.08 $_{\pm 0.27}$ |
| | ch-mod | x1.007 | 93.29 $_{\pm 0.17}$ | 90.05 $_{\pm 0.38}$ | 90.09 $_{\pm 0.18}$ | 92.11 $_{\pm 0.27}$ |
| | Ours | x1.32 | **95.76** $_{\pm 0.07}$ | **93.89** $_{\pm 0.29}$ | **93.81** $_{\pm 0.27}$ | **95.11** $_{\pm 0.10}$ |

In our architecture, both BU streams consist of the first part of the backbone and share their weights. The TD stream, unless specified otherwise, is a replica of the BU stream, in terms of layers structure and number of channels, combined with bilinear upsampling layers. The classifier is only attached to the BU2 stream. Information is passed between the BU1, TD and BU2 streams using lateral connections implemented as 1x1 convolutions. A task embedding layer (a fully connected layer) is added on the top of the TD stream. See an illustration of the full scheme in figure 1d and a detailed architecture description of the Multi-MNIST experiments in the supplementary materials.

**baselines** We compare our method both to a "single task" approach, where each task is independently solved and to a "uniform scaling" approach, where a uniformly weighted sum of the individual losses is being minimized. We have also compared our architecture to "ch-mod", a channel-wised vector modulation architecture (Zhao et al., 2018) and to a MOO (multi objective optimization approach) where the weights of loss items are dynamically tuned as suggested by Sener & Koltun (2018).

## 4.3 RESULTS

### 4.3.1 MULTIMNIST

We use the Multi-MNIST dataset to demonstrate our performance in case of uncorrelated tasks for 2, 3, and 4 tasks recognition problems with no additional hardware. All models trained using a standard LeNet architecture. We used a batch size of 512 images trained on 1 GPU with learning rate of $1e^{-3}$ using the Adam optimizer. Training curves are presented in figure [].

Figure 4b visualize the performance profile of the 2-classes experiment as a scatter plot of accuracies on task-LU and task-RL for the single task approach (vertical and horizontal lines correspondingly) and the multi-branched approach for several manually tuned loss weights (the blue dots). The scatter plot demonstrate a capacity problem, where better accuracies (above a certain limit) in one task cannot be achieved without being reflected as lower accuracies on another task. Our results are marked as a red star, showing better accuracies than the single-task case with much less parameters.

Table 2: Ablations on Multi-MNIST.

(a) spatial-wise image-aware modulation

| ALG | image | spatial | LU accuracy | RL accuracy | LL accuracy | RU accuracy | ↑ mean |
|---|---|---|---|---|---|---|---|
| ch-mod | × | × | 93.29 $\pm0.17$ | 90.05 $\pm0.38$ | 90.09 $\pm0.18$ | 92.11 $\pm0.27$ | 0 |
| – | √ | × | 93.52 $\pm0.18$ | 90.62 $\pm0.21$ | 90.64 $\pm0.18$ | 92.39 $\pm0.16$ | +0.41 |
| – | × | √ | 95.45 $\pm0.11$ | 93.29 $\pm0.17$ | 93.19 $\pm0.24$ | 94.60 $\pm0.15$ | +2.75 |
| Ours | √ | √ | **95.76** $\pm0.07$ | **93.89** $\pm0.29$ | **93.81** $\pm0.27$ | **95.11** $\pm0.10$ | +3.26 |

| (b) number of channels | | | | | (c) connectivity type | | | | (d) auxiliary losses | | |
|---|---|---|---|---|---|---|---|---|---|---|---|
| #ch | #P | LU ac | RL ac | | td,bu2 | LU ac | RL ac | | | LU ac | RL ac |
| dup | x1.29 | 97.09 | 96.11 | | +, + | 96.55 | 95.60 | | bu2 | 97.09 | 96.11 |
| 10 | x1.25 | 97.20 | 96.18 | | +, x | 97.06 $0.10$ | 96.15 $0.06$ | | bu2+bu1 | 96.75 | 95.71 |
| 6 | x1.10 | 96.88 | 96.05 | | +, ⊗ | **97.16** $0.05$ | **96.19** $0.14$ | | bu2+td | 96.82 | 95.56 |
| 2 | x1.02 | 96.59 | 95.55 | | x, ⊗ | 96.43 | 95.39 | | bu2+td+bu1 | 96.84 | 95.53 |
| 1 | x1.01 | 96.30 | 95.59 | | ⊗, ⊗ | 96.63 | 96.07 | | | | |

Table 1 summarizes our results on the Multi-MNIST experiment while sequentially enlarging the number of tasks. We show $mean_{\pm std}$ based on 5 experiments for each row. Our method achieves better results than the single-task baseline while using much less parameters (the third column shows the number of parameters as a multiplier of the number of parameter in a standard Lenet architecture). Other approaches, including the channel-wise modulation approach, achieve lower accuracy rates. Scaling the number of tasks keeps the accuracy gap almost without additional parameters.

### 4.3.2 ABLATIONS ON MULTI-MNIST

We further conducted ablation studies on Multi-MNIST, to examine several aspects of our proposed architecture. Table 2 shows the ablation results, analyzed as follows:

**Using spatial-wise and image-aware modulation modules**. Our experiments show that extending the existing channel-wise modulation architecture to an image-aware spatial-wise modulation architecture improves the results. Table 2a quantify the improvement in the results compared to the channel-wise modulation baseline (Zhao et al. (2018), first row in the table). We show $mean \pm std$ based on 5 repetitions of the full training pipeline for each row. Using a channel-wise image-aware modulation architecture by sequentially integrating information from the featuremaps in BU1 (second row) improves the accuracies by $\sim 0.4\%$. Using a spatial-wise modulation without using the information from BU1 stream (third row) improve the accuracies by $\sim 2.7\%$. Our approach, that uses both image-aware and spatial-wise modulation, improves the accuracies by a solid gap of $\sim 3.3\%$.

**Number of channels in the TD stream**. Table 2b compares the results accuracies of our proposed architecture (first line, where the TD stream is a replica of the BU stream which has 1, 10 and 20 channels in its feature-maps) with cheaper architectures which use a reduced number of channels along the layers in the TD stream. Our experiments show a trend line (the accuracies decrease when the number of channels in the TD stream decreases) and that optimizing the number of channels along the TD stream in terms of efficiency-accuracies tradeoff can be done (demonstrated by the second row in the table where higher accuracy achieved while using less parameters).

**Connectivity type**. Our architecture uses two sets of lateral connections; the first set passes information from the BU1 stream to the TD stream, and the second passes information from the TD stream to the BU2 stream. Table 2c compares the results accuracy of our proposed architecture when using different connectivity types to the TD stream (first column) and to the BU2 stream (second column). Here $+$ is an addition connectivity, $\times$ is a multiplication connectivity and $\otimes$ is a gated modulation with residual connection as described in Equation 3. The table shows higher accuracy when using addition connectivity along the TD stream and a small preference for the gated mod-

Table 3: Performance on CLEVR, higher is better. Our approach yields better accuracies also on correlated set of tasks with no additional hardware as tasks are added. Better accuracies are demonstrated both compared to the single task and uniform scaling approaches while using less parameters.

|  | ALG | $\#P$ | que1 accuracy | que2 accuracy | que3 accuracy | que4 accuracy |
|---|---|---|---|---|---|---|
| 2 tasks | Single task | x2 | 97.81 | 97.95 | | |
| | Uniform scaling | x1.5 | 98.09 | **98.19** | | |
| | ch-mod | x1.001 | 97.91 | 97.45 | | |
| | Ours | x1.56 | **98.17** | **98.19** | | |
| 3 tasks | Single task | x3 | 96.92 | 97.81 | **99.93** | |
| | Uniform scaling | x2 | 97.67 | 97.73 | 99.89 | |
| | ch-mod | x1.001 | 97.01 | 97.55 | 99.90 | |
| | Ours | x1.56 | **98.25** | **97.92** | **99.93** | |
| 4 tasks | Single task | x4 | 96.73 | 97.93 | **99.94** | 98.64 |
| | Uniform scaling | x2.5 | 97.47 | 97.93 | 99.92 | 98.64 |
| | ch-mod | x1.001 | 97.67 | 97.77 | 99.91 | 98.62 |
| | Ours | x1.56 | **98.43** | **98.16** | 99.93 | **98.66** |

ulation connectivity over the multiplication connectivity along the BU2 stream. To better compare between the two connectivity types we carried 5 experiments and report $mean \pm std$. We used the gated modulation connectivity type in all our experiments due to its slightly higher results.

**Auxiliary losses**. Our architecture, although usually uses only one classification loss at the end of the BU2 stream, can be easily adapted to integrate two auxiliary losses, one at the end of the BU1 stream (same classification loss) and the other on the image plane at the end of the TD stream (segmentation loss). Table 2d shows no additional improvement when using these auxiliary losses on the Multi-MNIST experiment. Note that a TD auxiliary segmentation loss (here, a binary cross entropy loss between the predicted digit and a zero-one map of the target digit) can also be used to add interpretability to our scheme. Examples are shown in the CUB200 experiment, section 4.3.4.

### 4.3.3 CLEVR

We used the CLEVR dataset to show our performance in case of correlated tasks (the questions on CLEVR are correlated) and to demonstrate our ability to enlarge the number of tasks with no extra hardware while keeping the targets accuracies.

Our results are summarized in Table 3. We trained all models using a VGG-11 architecture but decreased the number of channels in the output of the last convolutional layer from 512 to 128 to allow training with larger batch size. A detailed analysis of the number of parameters can be found in the supplementary materials. We used a batch size of 128 images trained on 2 GPUs with learning rate of $1e^{-4}$ using the Adam optimizer.

Table 3 shows that our results are better than both single task and uniform scaling approach while using much less parameters (the third columns shows the number of parameters of each architecture as a multiplier of the number of parameters in a single task VGG-11 backbone). Here, the channel-wise modulation approach uses the smallest number of parameters but also gets the worst results. The table also shows that scaling the number of tasks (with no additional hardware) is not only feasible but also may improve the results of each task separately. We further note that we used a TD layers that are a replica of the VGG-11 BU layers. Further reducing the number of parameters by decreasing the channel dimensions in the TD stream can be easily done but is not our main scope in this work.

Table 4: Performance on CUB200, higher is better. Our architecture is scalable with the number of tasks and outperforms other methods. All models trained for 200 epochs with lr 1e-4 using a resnet-18 backbone.

| | wing | uppertail | throat | nape | leg | eye | back | breast | forehead | belly | crown | bill | mean |
|---|---|---|---|---|---|---|---|---|---|---|---|---|---|
| Single task | 77.91 | 75.09 | 73.46 | 70.99 | 62.82 | 89.66 | 75.54 | 75.30 | 71.07 | 77.63 | 71.82 | 70.26 | 74.30 |
| Uniform scaling | 81.01 | 79.46 | 77.55 | 75.94 | 64.57 | 90.40 | 79.46 | 78.67 | 74.58 | 80.62 | 75.25 | 72.07 | 77.46 |
| MOO | 82.78 | 82.17 | 77.37 | 75.66 | 64.84 | 91.39 | 81.15 | 79.63 | 75.03 | 81.57 | 75.44 | 73.80 | 78.40 |
| ch-mod | 82.07 | 81.72 | 80.03 | 77.20 | 68.61 | 91.30 | 81.29 | 81.22 | 76.91 | 82.29 | 77.87 | 76.23 | 79.72 |
| ch-mod + loc | 78.84 | 76.79 | 75.18 | 72.09 | 58.75 | 90.35 | 75.99 | 77.17 | 71.30 | 78.74 | 72.54 | 67.07 | 74.61 |
| Ours + loc | 84.90 | 81.77 | 81.01 | 79.53 | 67.60 | 91.08 | 83.22 | 83.64 | 78.55 | 84.41 | 79.96 | 75.61 | 80.94 |

### 4.3.4 CUB 200

We used the CUB-200 dataset to further demonstrate our performance on correlated tasks in real-world images, scaling the number of tasks and using another type of backbone architecture (a Resnet backbone). In contrast to previous experiments, we did not aim at reducing the number of parameters (since we are using a Resnet backbone); rather we demonstrate better performance, and our built-in capability to visualize the attention maps at the end of the TD stream.

We trained all models using a Resnet-18 architecture. We used a batch size of 128 images trained on 2 GPUs with learning rate of $1e^{-4}$ using the Adam optimizer for 200 epochs. While training our architecture we add an auxiliary loss at the end of the TD stream. The target in this case is a one-hot 224x224 mask, where only a single pixel is labeled as foreground, blurred by a Gaussian kernel with a standard deviation of 3 pixels. Training one task at a time, we minimize the cross-entropy loss over the 224x224 image at the end of the TD softmax output (which encourages a small detected area) for each visible ground-truth annotated task/part. For a fair comparison, we also compared our results to the channel-wised modulation architecture trained with the same localization auxiliary loss (on the coarse map at the end of the BU2 stream, fifth line in the table).

Figures 4c and 4d demonstrate the attention maps produced by our architecture in inference time. Figure 4c is an example where the predicted mask is well localized on the crown of the bird (the task) and the color is correctly predicted. Figure 4d demonstrate an error case where the breast of the bird is not well localized by the mask and as a consequence the color is wrongly predicted. More examples of interest are shown in the supplementary materials.

Our quantitative results are summarized in Table 4. The results show better accuracy of our scheme compared to all baselines. We specifically show better accuracy compared to the channel-wise modulation scheme, indicating the preference of our image-dependent spatial-wise modulation process on the CUB200 database.

## 5 SUMMARY

We proposed a novel architecture for multi-task learning using a top-down modulation network. Compared with current approaches, our scheme does not use task-dependent branches or task-dependent modules, and the modulation process is executed spatial-wise as well as channel-wise, guided by the task and by the information from the image itself. We tested our network on three different datasets, achieving on par or better accuracies on both correlated and uncorrelated sets of tasks. We have also demonstrated inherent advantages of our scheme: adding tasks with no extra hardware that result in a decrease in the total number of parameters while scaling the number of tasks, and allowing interpretability by pointing to relevant image locations.

More generally, multiple-task learning algorithms are likely to become increasingly relevant, since general vision systems need to deal with a broad range of tasks, and executing them efficiently in a single network is still an open problem. In future work we plan to adapt our described architecture to a wider range of applications (e.g. segmentation, images generation) and examine possible combinations of approaches such as combining partial-branching strategy with our TD approach. We also plan to study additional aspects of multi-task learning such as scaling the number of tasks and tackling the catastrophic forgetting problem.

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

Table 5: Number of parameters in the M-MNIST and CLEVR architectures

(a) M-MNIST experiment

| Module / Architecture | # params |
|---|---|
| Recognition backbone | 21,250 |
| Each branch | 3,000 |
| TD with laterals | 6,651 |
| task embedding | 320 |
| Single-Task architecture | 24,250 |
| Multi-branched architecture | $21,250 + 3000 \cdot K$ |
| TD modulation architecture | $21,250 + 9651 + 320 \cdot K$ |

(b) CLEVR experiment

| Module / Architecture | # params |
|---|---|
| Recognition backbone | 7,448,256 |
| Each branch | 7,473,152 |
| TD with laterals | 8,306,688 |
| task embedding | 1568 |
| Single-Task architecture | 14,921,408 |
| Multi-branched architecture | $14,921,408 + 7,473,152 \cdot K$ |
| TD modulation architecture | $14,921,408 + 8,306,688 + 1568 \cdot K$ |

## A  IMPLEMENTATION DETAILS

For the MultiMNIST experiments, we use an architecture based on LeNet (LeCun et al., 1998)). We followed Sener & Koltun (2018) and use two 5x5 convolutional layers and one fully-connected layer as the shared backbone and two other fully-connected layers as task specific branches for the multi-branched architecture (See figure 5a for details).

Our architecture is illustrated in figure 5b. We use the shared backbone and a single branch as the recognition network (BU2). On the TD stream we use an embedding layer followed by two 5x5 convolutional layers. BU1 share the same weights with BU2. the three subnetworks are combined together using lateral connections, implemented as 1x1 convolutions. The networks for the CLEVR and CUB200 experiments where similarly implemented using VGG-11 and ResNet-18 backbones correspondently. The exact number of parameters used by the Multi-MNIST and by the CLEVR architectures are summarized in table 5.

## B  ADDITIONAL QUALITATIVE EXAMPLES ON CUB200

To demonstrate our interpretability capabilities we trained our proposed network with an auxiliary localization cross entropy loss on the last layer of the TD stream (details in section 4.3.4). Here we present several more examples of interest we did not include in the main text.

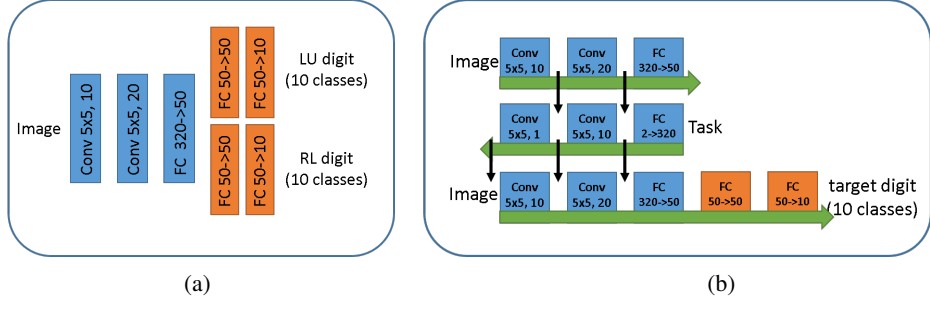

(a)                    (b)

Figure 5: Architectures used in the two-classes multi-MNIST experiment: (a) Multi-branched architecture (b) Ours top-down modulation architecture. The number of parameters in each architecture are summarized in table 5a.

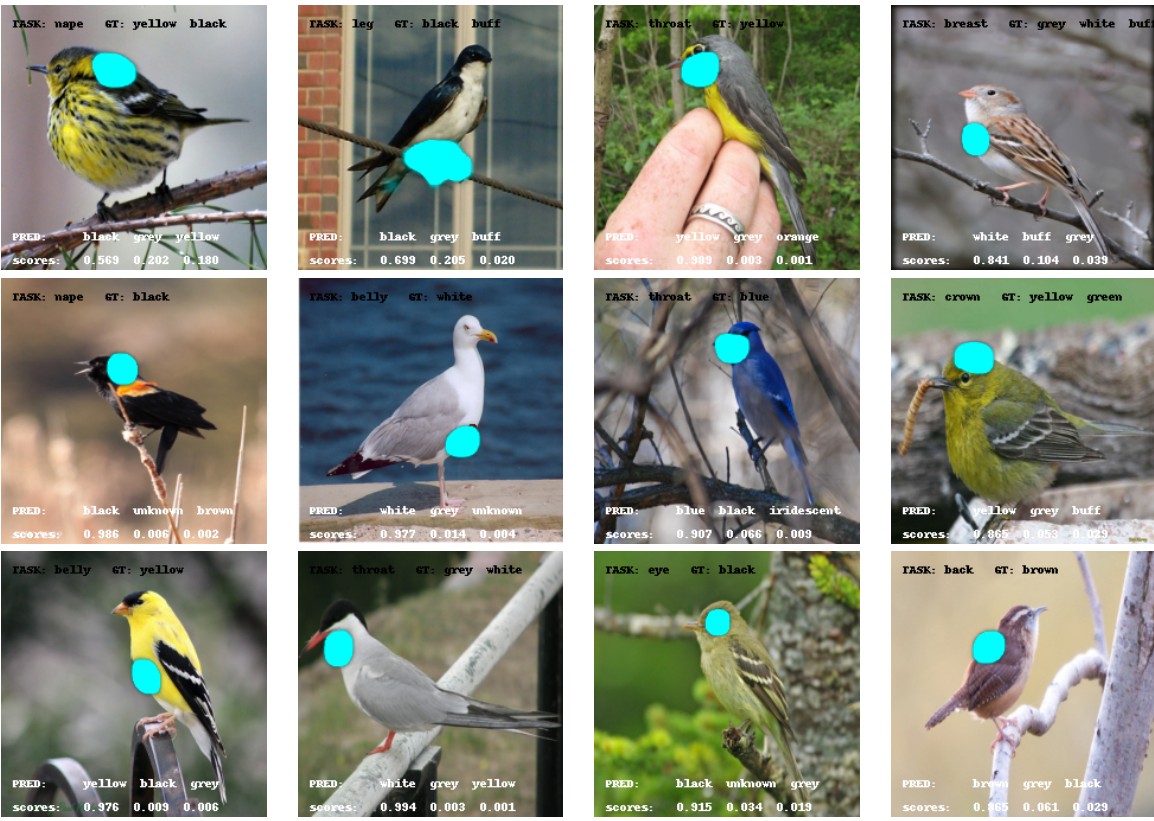

Figure 6: More qualitative examples to demonstrate our ability to identify the relevant regions that most affected the network prediction. In all of these images the target part (the task, shown in the upper part of each image), is precisely localized and the prediction (shown in the lower part of each image) follows the ground truth. Best viewed in color while zoomed in.

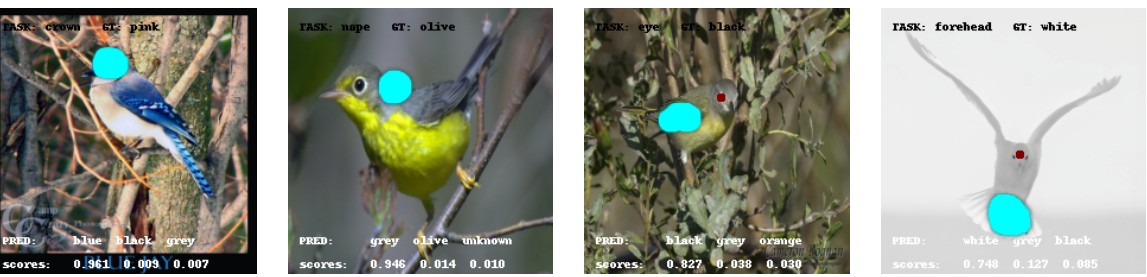

Figure 7: Error cases. Left images demonstrate good examples, counted as failure cases due to annotations errors. Our network successfully localize the asked part and correctly predict its color. Right images demonstrate bad localization examples. Ground truth classes were still predicted, with a very high score, maybe due to the correlated nature of the tasks. Best viewed in color while zoomed in.

