# OpenReview forum: "Task-Based Top-Down Modulation Network for Multi-Task-Learning Applications"
_ICLR.cc/2020/Conference — Reject_

### Official Review · AnonReviewer3 · 2019-10-25
**Official Blind Review #3**

**Rating:** 3

**Review:**

This paper tackles the multi-task setting by using modulation connections between three network pipelines, i.e., one bottom-up network to contextualize the problem, one top-bottom network that is conditioned on the tasks, on a last bottom-up network that solves the task.
The key contribution of the paper is to introduce a feature-wise and spatial-wise tensor to modulate the different neural pipelines better. Finally, they assess the proposed method on three datasets: Multi-MNIST, a yes/no CLEVR, and CUB-200.

The abstract, introduction, and related works are pretty clear. Figure 1 is also a nice summary that puts the paper architecture in perspective with another approach, and it is a very insightful sketch. I appreciate the effort of the authors to release the code with several baselines. I also acknowledge the diversity of tasks that are studied.

However, I have three concerns that I am willing with the authors.

My first concern deals with the method description, which I found a bit misleading. Thus, I not sure that I fully got all the subtleties of the proposed method. The mathematical notation is misleading: Are Y, and X function of (x,y,ch) or are tensors over x, y, z. Later in the text, W is defined over (ch, t), but it is also mentioned that t is an input. Thus, is W \in R^(CxT) or W(t) \in R^(C). Besides, the implicit tilling with * makes things even harder to follow. On a different side, what do you mean by training the convolution network instead of optimizing W. Is W fixed? Do you use simply the feature map after 1x1 conv as mentioned in 4.2? How do you embed the task t in general, how do you append it to the feature map of BU1.
In the current paper state, I would not be able to reproduce the experiments.

The second concern relies on the results. The gap between the methods is tiny, e.g. max 0.5 in 2-MNIST, and may fluctuate a lot from one experiment to another, e.g., it is weird that 3-MNIST is harder than 4-MNIST. Note that the same observations can be applied to the CLEVR. Therefore, it is hard to assess the method without the std over at least 5 seeds. The result only convinces me regarding 4-MNIST without such std.

My third concern is about CUB200 experiments. The authors used an auxiliary loss on top of TD to help to visualize the network decision. As such auxiliary losses provide additional information, I have the following question: did you add the same losses to other baselines? Did you use a stop-gradient before decoding the feature-maps? Otherwise, the comparison between methods may not be fair


Remarks:
 - I am missing some results from external literature. For instance, even if I prefer your setting CUB300  over 312 questions, it would have be nice to add such experiments in the appendix. (or literature baselines on N-MNIST)
 - Please report the original MOO too results in addition to your experiments
 - Why ch-mod is missing in 4-CLEVR?
 - Can you describe how did you pick the CLEVR questions (before/after computing the results)? It would have been nice to have experiments that randomly pick K questions (and repeat the process N time + report std), or even dynamically condition on the question at hand.
 - it took me quite some time to understand the meaning of #P, please make it explicit from the beginning, or add in the caption!
 - Although releasing the code is good, I also encourage you to put a table in the appendix with the hyper-parameters. The paper should be as much self-content as possible. It is also hard to evaluate the quality of the training time during the review
 - typo: extra parenthesis in Eq 4

In conclusion, the authors give some good intuition about promising methods, but I had some difficulties in understanding all the details of their approaches. Besides, I am missing both std and external references to assess the quality of the methods. In this current state, I cannot recommend paper acceptance even if I acknowledge several qualities of the paper, but I am open to discussion.



**Experience Assessment:**

I have published one or two papers in this area.

**Review Assessment: Checking Correctness Of Derivations And Theory:**

N/A

**Review Assessment: Checking Correctness Of Experiments:**

I assessed the sensibility of the experiments.

**Review Assessment: Thoroughness In Paper Reading:**

I read the paper at least twice and used my best judgement in assessing the paper.

---

> ### Author Response · Authors · 2019-11-13
> **Discussion of review #3**
>
> First of all, thanks for the helpful review.
>
> We have updated the article and uploaded a revised version based on the reviews,
> Specifically we re-wrote section 3 (Approach). We find the explanations now clearer.
>
> One of your major remarks is that it’s better to assess the quality of the method based on mean+std over several seeds:
> We have repeated all our training in the Multi-MNIST experiment 4 more times (to get 5 separate training and evaluation pipeline to every row in the table) and report mean + std, tables 1+2, pages 7-8 in the experiment section).
>
> Regarding the gap between  the 3-MNIST exp and 4-MNIST exp:
> Several M-MNIST images are demonstrated in figure 2 in the article. The digits are i.i.d. between images and between tasks (uncorrelated). We believe that the gap between the experiments is because the different overlap areas of the digits in those experiments which carry useful information for the classification process.
>
> You are right that the use of the localization auxiliary loss provides additional information, however:
> a.	We also run the channel-wise modulation architecture on the CUB200 with the same auxiliary loss (on the top of the BU2 stream). The results were worse than our architecture. This experiment is now added to the article, table 4 page 10. Thank you for arising this point.
> b. Regardless of performance, our goal by using the auxiliary loss was to demonstrate interpretability.
>
> The ch-mod is now added to the CLEVR experiment.
>
> The questions in the CLEVR experiment were arbitrarily chosen, simply as it sound. Your suggestions are correct, but cannot be applied during the review time.

---

> > ### Comment · AnonReviewer3 · 2019-11-15
> > **Response**
> >
> > Thank you for your concise and clear responses. Several concerns are correctly answered, and they would enrich the discussion. I particularly appreciated the std. I would have also loved to add your method without location in Tab 4, as it would be proof that your model does not merely leverage better this information that other approaches. In this end, I am a bit more leaning toward acceptance, and I have more arguments for the upcoming discussions for sure.
> >
> > Remarks:
> >  - I would recommend the authors to cite the following papers:
> >      * Tenenbaum, Joshua B., and William T. Freeman. "Separating style and content with bilinear models." Neural computation 12.6 (2000): 1247-1283. -> Your method is a form of bilinear model.
> >      * Dumoulin, Vincent, Jonathon Shlens, and Manjunath Kudlur. "A learned representation for artistic style." arXiv preprint arXiv:1610.07629 (2016).  -> As the method also use modulation per task/style
> >      * https://distill.pub/2018/feature-wise-transformations/ -> The authors formalize the modulation principle which is used in this paper

---

### Official Review · AnonReviewer1 · 2019-11-05
**Official Blind Review #1**

**Rating:** 3

**Review:**

This paper introduces a new light-weight framework for multi-task learning. In this method, the combination of extracted image features and task are fed into a top-down network which is responsible for generating a task-specific weight matrix. The weights are next convolved with the input image as an input to the task-agnostic bottom-up network that generates the labels.

The idea of the paper interesting. The main shortcoming of the paper in my point of view is that all the numbers are reported as a single number, so they are prone to be changed by using different initial networks or optimizers. Here are some more comments:

1) One limitation of the result section is that all the numbers are reported as static numbers. I am interested to see the training curves, either in using the wall-clock time or iteration in the x-axis and testing accuracy in y.

2) Sections 3.1 and  3.2 as the main parts are not well-written. The shapes of the tensors are vague. What is the y,x in the parentheses? What does ch stand for? (defined?) I think that this part of the paper requires significant improvement.

3) One valid question is how the proposed method is scalable. For example, can a model trained for 3 tasks used for 4 tasks? How hard is adding a new task? Also, worths comparing it with the learning from scratch.

4) In Section 3, the discussion about the loss function is missing. I believe that the explanation of how to choose a loss function as well as auxiliary losses should be move there. Also, I didn't find the current explanation of BU1 and TD auxiliary loss for Multi-MNIST very clear.

5) Why the results of your method is better than the single model? This behavior should be justified. My impression is that each task trained independently should outperform any multi-training method. Your results seem counter-intuitive in this respect.

6) I am not able to make any strong conclusions from Section 4.3.2. It is really hard to tell which connection is better based on a single number. I would suggest providing confidence intervals for making such kind of arguments. For example, you may train from 10 different network initializations and use them to construct more reliable estimates. I also believe that more reliable estimations are required for Table 3.


Minor:
* In paragraph 2 of pages 2, you mention "as illustrated in Figure 2a". I do not see the attention to a part of the image. Am I missing something? A similar issue exists in the next sentence: I don't see any content-related modulization in Figures 2b and 2c. Please clarify.
* use comma after equations if the equations are not ending the sentences. For example, add a comma after eq (1), (2) and (3). Also on page 4, "Where $W$" -> "where $W$".
* Page 4, "Our method address" -> "Our model addresses"
* Where the third column of Table 1 is defined? On page 8. Move it to earlier sections.
* In Table 2b, you have used +x, but the notation for gated modulation is something else in the text.
* Are LL and RU used in Table 1 defined in the text?
* The bold numbers in Figure 2b seem wrong. If you are bolding the large accuracies, be consistent in all tables.
* Font of table 4 can be larger

**Experience Assessment:**

I do not know much about this area.

**Review Assessment: Checking Correctness Of Derivations And Theory:**

I assessed the sensibility of the derivations and theory.

**Review Assessment: Checking Correctness Of Experiments:**

I assessed the sensibility of the experiments.

**Review Assessment: Thoroughness In Paper Reading:**

I read the paper at least twice and used my best judgement in assessing the paper.

---

> ### Author Response · Authors · 2019-11-13
> **Discussion of review #1**
>
> First of all, thanks for the helpful review.
>
> One of your major remarks was that all of our numbers were reported as single numbers:
> We have repeated all our training in the Multi-MNIST experiment 4 more times (5 separate training and evaluation pipelines to every row in the table) and report the mean + std in the tables (tables 1+2 in the experiment section, pages 7-8). We have also added training curves, figure 4b, page 6.
>
> The approach section was rewritten, now better explains our modulation scheme and main contributions.
>
> Regarding your question why our method is better than a single model – when the tasks are correlated (CLEVR /  CUB200) we expect our method to be better since the gradients from other tasks pass relevant information to any specific task. When the tasks are uncorrelated (MMNIST) the better results may be attributed to our spatial-wise modulation, focusing on the relevant digit. Table 2a, page 8, emphasize our contributions in this aspect.
>
> All the minor suggestions have been addressed.
>
> We have updated the article and uploaded a revised version based on the reviews.

---

> > ### Comment · AnonReviewer1 · 2019-11-15
> > **Most of the comments are addressed, but still there are some doubts**
> >
> > In the revised version of the paper, the authors have addressed some of my comments that have improved the quality of the paper.
> >
> > However, I am still not convinced about the justification of comment  5. Of course, each particular task is the most correlated task to itself, so the single model should always outperform. One of my most important concerns were 6 that has not been addressed at all.
> >
> > I know that due to the limited time, it was not possible to address my other comments. At this point, I am neutral about this paper.

---

### Official Review · AnonReviewer2 · 2019-11-05
**Official Blind Review #2**

**Rating:** 3

**Review:**

> What is the specific question/problem tackled by the paper?
This paper tackles a restricted multi-task setting where the task is known. The main contribution is a new architecture for training a task conditional model. The new architecture is reminiscent of an encoder-decoder-classifier with ladder (latent) connections, the decoder is conditioned on task ID. The claim is this is a type of modulation, it is unclear. Results on three multi-task datasets show that the proposed method is slightly better than compared methods and single task learning. There is no theory or loss function to analyze.

> Is the approach well motivated, including being well-placed in the literature?
In my opinion, this is lacking. The assumption that task ID is known is fairly severe. Unfortunately the prior works cited also have this restriction, whereas few papers under the topic of continual learning have removed this limitation. This assumption/drawback needs to be clearly mentioned in the paper and discussed if it is realistic? A related shortcoming is that the training data simultaneously comes from all the tasks, whereas prior work has looked at the more interesting setup where tasks arrive sequentially and incrementally.
- Reference [1] seems relevant and should be cited as it shows context dependent gating of tasks / modulation as well. Other missing references e.g. learning without forgetting (LwF) [2] and [3].
- There is not a clear explanation to think that this is modulation since the result is only passed through a residual connection. More importantly there is no discussion on these important issues. I found the writing to be brief and sketched.
- in the introduction, it would be good to define multi-task learning with the assumption made clear. It would be good to introduce what you mean by TD and BU clearly
- Another drawback is assuming the tasks being encoded as integers, whereas there might be a continuous task space with interpolation, or hierarchical task structure.
- "However, all of these works modulate the recognition network
channel-wise, using the same modulation vector for all the spatial dimension of the feature-maps." - why is this not enough? A nontrivial explanation or discussion is needed. Simply extending to W(t, y, x, ch) would increase performance by a little.
- how is the proposed model different from a conditional model like a task conditional classifier? Also in experiments.
- How is the proposed model different from an encoder-decoder? The impact of "modulation" is not clear.
- "We can scale the number of tasks with no additional layers." - task conditional classifier can also scale in this way to the number of tasks. This claim is not valid.
- Page 3: "uncorrelated gradients from the different tasks" - need not be uncorrelated, but still can be interfering
- next about Kendall (2018) and Sener (2018) - need to compare and contrast to them.
- Last para on page 3 seems not relevant.
- Modulation equations: this seems specific to CNNs. How would you extend this technique to beyond CNNs to recurrent units or even simpler MLPs? Modulation as a technique has been successfully applied in these architectures as well.
- "added to the input tensor X through a residual connection" - this is not clear at all. Are the residual connections not shown in Fig 1(d)?
- "it to be unfeasible due to their large dimensions" - can you explain please? later you say "To avoid the unfeasible computation burden of directly optimizing W"
- Fig 1d, would be good to mark the modulation arrows in a different color

> Does the paper support the claims? This includes determining if results, whether theoretical or empirical, are correct and if they are scientifically rigorous.
Having said that, I like the experimental section even in the restricted setup. But a lot of details are missing. It is not surprising that there is slight increase in performance over channel modulation due to the increase expressivity.
- table 1: why is there degradation in the LL task across all methods? the introduction of an additional task seems to bring the performance back up. It seems to be a weakness of your method. Please improve the discussion. I'm inclined to think that the tasks are not uncorrelated, as claimed by the authors.
- table 1: how did you arrive at the number of parameters like 1.12x? Doesn't the separate BU and TD nets mean you have at least 2x parameters compared to single task? It seems the larger number of params in single task is mainly coming from the hidden layers?
- table 1: it would be fair for the comparison methods to have equal number of parameters as the proposed method.
- Missing experimental comparison to Kendall et. al. 2018
- Missing details about reproduction of results from Sener (2018)
- An important baseline would be to show image sensitive full tensor modulation without the new architecture. Similar to XdG.
- Another baseline should be a task-conditional classifier that takes task as input along with the image.
- ablation study: what are the auxiliary losses? I could not find any details.
- The third experiment with CUB seems to use a different loss function that the other methods. This is somewhat hard to evaluate.
- number of parameters are not reported for the CUB experiment
- "where only a single pixel is labeled as foreground, blurred by a Gaussian kernel" needs more details about the smoothing
- In the CUB experiment, due to lateral connections, the top-down result 224x224 image is not the only input to the BU2 classifier, the interpretability argument is weak.
- Why did you choose these 4 questions from CLEVR? There are many interesting types of questions that can be handled.



**Experience Assessment:**

I have published in this field for several years.

**Review Assessment: Checking Correctness Of Derivations And Theory:**

N/A

**Review Assessment: Checking Correctness Of Experiments:**

I carefully checked the experiments.

**Review Assessment: Thoroughness In Paper Reading:**

I read the paper thoroughly.

---

> ### Author Response · Authors · 2019-11-13
> **Discussion of Review #2**
>
> First of all, thanks for the thoughtful review.
>
> Regarding your general concern about the scope of the paper (known task-ID is a severe condition, continual learning is more interesting).
> We agree that there are other realistic setting but we also follow an interesting, realistic and an active line of research. References [1], [2], [3], [4] all uses a setting similar to ours (modulation networks with a known task-ID). Reference [4] is interesting in this aspect as it applies channel-wise modulation networks with a given task-ID to continual learning applications. We, on the other hand, continue this line of research by extending channel-wise modulation tensor-modulation ...
>
> [1]  A Modulation Module for Multi-task Learning with Applications in Image Retrieval, ECCV 2018
> [2]  Attentive Single-Tasking of Multiple Tasks, CVPR 2019
> [3] Many Task Learning with Task Routing, arxiv preprint
> [4] Superposition of many models into one, arxiv preprint
>
> Request - you gave several reference numbers in the review, but the references were not listed. can you add them?
>
> We next go through specific comments:
>
> A major remark was to justify and quantify our main technical contribution (adding 1. spatial-wise and 2, image-aware modulation to the existed channel-wise modulation architectures. We motivated our contribution in the introduction (page 2, paragraph 2) and now added to specifically quantify the contribution of each of our additions, summarized in table 2a, section 4.3.2.
>
> We rewrote section 3 (Approach) to better explain our modulation scheme. We treat it as a modulation because adding the residual connections change x*w to x*(w+1). Following your question, we have also explained why a naive extension of [1] is infeasible (briefly, the number of parameters to optimize is to large and depend on H, W and the number of tasks - it cannot be applied to large images and is not scalable with the number of tasks).
>
> We added an appendix with a calculation of the number of parameters. Briefly, for K tasks – a single task networks use one separate network for each task (K times the number of parameters found in a single base network), Multi-branched architectures add several branches on a common backbone (one backbone + the number of parameters within a branch times K), our architecture uses only one branch and our TD usually has much less parameters than the BU backbone (because if a convolutional stage uses a series of convolutions like (256->512, 512->512) we use (512->256, 256->256)).
>
> We reproduced the results of Sener(2018) from their official github code.
>
> Regarding the reduced performance on the LL digit:
> Several M-MNIST images are demonstrated in figure 2 in the article. The digits are i.i.d. between images and between tasks (uncorrelated). We believe that the degradation of the LL task accuracy is because the specific overlap areas of this task/location carry useful information for the classification process.
>
> We have updated the article and uploaded a revised version based on the reviews.

---

### Author Response · Authors · 2019-11-13
**Task-Based Top-Down Modulation Network for Multi-Task-Learning Applications**

Thanks for the helpful and detailed reviews,

We have updated the article and uploaded a revised version based on the reviews.

Our main changes are:
1. We specifically tested and quantified our main technical contributions, adding 1. spatial-wise and 2, image-aware modulation to the existed channel-wise modulation architectures (table 2a, section 4.3.2.).
2. We run the Multi-MNIST experiments 4 more times (to get 5 repetitions of the full training and evaluation pipelines) and report mean+std, tables 1+2, pages 7-8.
3. We re-wrote the Approach section, making it clearer.

---

### Decision · Program_Chairs · 2019-12-19

**Decision:**

Reject

**Comment:**

The paper is interested in multi-task learning. It introduces a new architecture which condition the model in a particular manner: images features and task ID features are fed to a top-down network which generates task-specific weights, which are then used in a bottom-up network to produce final labels. The paper is experimental, and the contribution rather incremental, considering existing work in the area. Experimental section is currently not convincing enough, given marginal improvements over existing approaches - multiple runs as well as confidence intervals would help in that respect.